# Parental effects provide an opportunity for coral resilience following major bleaching events

**Elizabeth A. Lenz**[1], **Megan J. Donahue**[2], **Ruth D. Gates**[2†], **Hollie M. Putnam**[3], **Eveline van der Steeg**[4], **Jacqueline L. Padilla-Gamiño**[5]*

1 University of Hawai'i Sea Grant College Program, University of Hawai'i at Mānoa, Honolulu, HI, United States of America, 2 Hawai'i Institute of Marine Biology, University of Hawai'i at Mānoa, Kāne'ohe, HI, United States of America, 3 Department of Biological Science, University of Rhode Island, Kingston, Kingston, RI, United States of America, 4 School of Natural and Environmental Science, Newcastle University, Newcastle, United Kingdom, 5 School of Aquatic and Fishery Sciences, University of Washington, Seattle, WA, United States of America

† Deceased.

* jpgamino@uw.edu

**Data Availability Statement:** The data is currently held in a public repository on GitHub: https://github.com/ealenz/Mcap-BNB-Reproduction-SelectiveBreeding.

## Abstract

Identifying processes that promote coral reef recovery and resilience is crucial as ocean warming becomes more frequent and severe. Sexual reproduction is essential for the replenishment of coral populations and maintenance of genetic diversity; however, the ability for corals to reproduce may be impaired by marine heatwaves that cause coral bleaching. In 2014 and 2015, the Hawaiian Islands experienced coral bleaching with differential bleaching susceptibility in the species *Montipora capitata*, a dominant reef-building coral in the region. We tested the hypothesis that coral bleaching resistance enhances reproductive capacity and offspring performance by examining the reproductive biology of colonies that bleached and recovered (B) and colonies that did not bleach (NB) in 2015 in the subsequent spawning seasons. The proportion of colonies that spawned was higher in 2016 than in 2017. Regardless of parental bleaching history, we found eggs with higher abnormality and bundles with fewer eggs in 2016 than 2017. While reproductive output was similar between B and NB colonies in 2016, survivorship of offspring that year were significantly influenced by the parental bleaching history (egg donor × sperm donor: B × B, B × NB, NB × B, and NB × NB). Offspring produced by NB egg donors had the highest survivorship, while offspring from previously bleached colonies had the lowest survivorship, highlighting the negative effects of bleaching on parental investment and offspring performance. While sexual reproduction continues in *M. capitata* post-bleaching, gametes are differentially impacted by recovery time following a bleaching event and by parental bleaching resistance. Our results demonstrate the importance of identifying bleaching resistant individuals during and after heating events. This study further highlights the significance of maternal effects through potential egg provisioning for offspring survivorship and provides a baseline for human-assisted intervention (i.e., selective breeding) to mitigate the effects of climate change on coral reefs.

**Funding:** This paper is funded in part by a grant from the National Oceanic and Atmospheric Administration, Project A/AS-1; which is sponsored by the University of Hawai'i Sea Grant College Program, SOEST, under Institutional Grant No. NA22OAR4170108 from NOAA Office of Sea Grant, Department of Commerce. The views expressed herein are those of the author(s) and do not necessarily reflect the views of NOAA or any of its subagencies. UNIHI-SEAGRANT-4941.

**Competing interests:** The authors have declared that no competing interests exist.

## Introduction

Ocean warming caused by anthropogenic greenhouse gas emissions is one of the primary threats to the function of shallow tropical coral reefs [1,2]. Prolonged warming above the local thermal threshold for bleaching coupled with high irradiances can cause severe coral bleaching [3], the disruption of the nutritional symbiosis between the coral host and its unicellular dino-flagellates, Symbiodiniaceae (formerly, *Symbiodinium* spp.) [4]. This can subsequently result in increased rates of disease transmission [5] and mortality [6] along with reduced calcification rates and reproductive capacity in corals [7,8]. Continual declines in coral cover are predicted given the range of local and global disturbances simultaneously acting on coral reefs, with warming ranked as the most severe [9–11]. Identifying sources of resilience in coral reef eco-systems, such as locating exceptional coral genotypes that can thrive under extreme warming or temperature fluctuations, will be key in maintaining and restoring reefs for the future.

Differential bleaching susceptibility [12–14] during a thermal stress event illustrates biologi-cal variation within populations that may serve as a source of resilience and an opportunity for selection through reproductive success [15,16]. Thermal tolerance and capacity to recover after bleaching are important factors that influence sexual reproduction, recruitment, and suc-cess of future generations to adapt [7,8,17,18]. Successful sexual reproduction and recruitment are essential in maintaining coral populations [19], repopulating disturbed coral reefs [20–23], and enhancing genetic diversity within populations to overcome selective pressures [24,25]. However, parental investment in gametogenesis is energetically costly [26] and for corals reproductive cycles may exceed six to ten months [27,28]. Therefore, prolonged environmental stress can drive prioritization of energetic investment into basic metabolic function and repair, at the expense of growth and sexual reproduction [29–31]. Importantly, this tradeoff in ener-getic investment is likely to depend on the susceptibility and severity of coral bleaching, with greater energy available for reproduction in corals resistant to bleaching [32].

Previous studies have identified some of the way coral bleaching can impact aspects of sex-ual reproduction [8,33] and dampen recruitment [34,35]. For example, after the 1987 coral bleaching event in the Caribbean, *Orbicella annularis* recovered from bleaching by metaboliz-ing tissue biomass, but did not complete gametogenesis in the following months, whereas colo-nies that had not bleached of the same species were able to develop and release gametes [7]. Similarly, during the 1998 bleaching event on the Great Barrier Reef, bleached corals showed high variation in reproduction compared to colonies resistant to bleaching nearby that experi-enced the same thermal stress. For acroporid species, reproductive polyps were more common in colonies that did not bleach, with larger eggs at higher densities per polyp than colonies that bleached and recovered [30]. More resolution is needed to better understand the impact and extent of coral bleaching events on the early life cycles of coral, from the stress event through recruitment.

Given logistical complexities and challenges, most studies have primarily investigated gametogenesis in the life cycle of coral with some understanding of cross-generational effects (i.e., parental, carry-over, or transgenerational effects) following major bleaching events. The impacts of coral bleaching may last for months to years after the initial thermal stress [36], and can manifest in life stages downstream such as fertilization [37–39], larval development, and recruitment [34,35,40]. Between the 2005 and 2010 bleaching events in Panama, Levitan et al. (2014) found that thermally tolerant *Orbicella franksi* recovered the capacity to produce and release gametes more quickly (within 3 to 5 years) than the more thermally sensitive *O. annu-laris*. While these studies demonstrate a range of responses in sexual reproductive biology and ecology during recovery post bleaching (i.e., gametogenesis and recruitment), few studies have followed both the intra- and intergenerational impacts of bleaching. Recent marine heatwaves

eliciting differential coral bleaching of *M. capitata* in Hawaiʻi provide an opportunity to compare the impacts of parental bleaching history on coral reproduction and offspring performance during recovery and offer potential insight on coral resilience [15,41,42].

Coral reefs in the subtropical waters of Hawaiʻi were largely naive to global bleaching events [43–45] with bleaching events first recorded in the Main Hawaiian Islands in 1996 and then in the Northwestern Hawaiian Islands in 2002 [43–45]. However, the Hawaiian Archipelago experienced "the blob" heatwave, followed by an El Niño that resulted in severe back-to-back coral bleaching in 2014 and 2015 (Fig 1A) [46,47]. During these consecutive bleaching events, degree heating weeks (DHW) in the Main Hawaiian Islands exceeded 8 weeks by September in both years [46,47]. In Kāneʻohe Bay (Oʻahu, Hawaiʻi), ~70% of corals on the shallow reefs (< 2 m depth) bleached and exhibited 13–22% mortality in 2014 and 2015 [46,48–50]. During both events in Kāneʻohe Bay, colonies of the dominant reef-building coral, *Montipora capitata*, visibly bleached or remained pigmented during prolonged heat stress (Fig 1B). Despite widespread bleaching, approximately 70% of *M. capitata* that bleached in 2014 and 2015 were considered recovered by the following December and January based on visual coloration [12,14,15,46,51–53].

*M. capitata* demonstrates relatively high tolerance against multiple local and global stressors [54,55], with varied sensitivity among individual colonies and their traits measured under elevated temperature [15,51], such as survivorship [49], growth [45], and biomass composition [45,53,54,56–58]. Reproductive effort of *M. capitata*, particularly oocyte characteristics and spawning, has shown little response to warming [36,59]. This reproductive response may have contributed to its ecological success along the fringing and patch reefs of Kāneʻohe Bay in the past. However, percent of motile sperm from *M. capitata* declined from 80–90% in 2011 to 40.5% in 2015, corresponding with the consecutive bleaching events in Kāneʻohe Bay [36]. For *M. capitata*, oogenesis can begin as early as July, which means that early egg development may cooccur with severe, prolonged warming events (July-October), and later egg development continues when corals are recovering from these events (November-August). This could create a strain on energetic resources when corals are compromised during a substantial fraction of the typical gametogenic cycle [60,61]. Therefore, tracking *M. capitata* through subsequent spawning seasons after bleaching events can reveal the reproductive capacity of this species as ocean temperature continues to increase.

In this study, we examined cross-generation plasticity (i.e., parental effects) to determine how parental response to environmental events influence reproduction [62]. We measured the reproductive biology of *M. capitata* for two spawning seasons (2016 and 2017) following bleaching events (2014 and 2015). We tested the following hypotheses: (i) that parental bleaching history [bleached (B) and nonbleached (NB)] would affect reproductive performance in subsequent spawning seasons and (ii) intentional crosses of gametes from parent colonies of differential bleaching history would influence offspring success (Fig 2A). In 2016, we tested the second hypothesis and quantified the downstream effects of parental bleaching history from gamete release to settlement of the offspring in parent colonies that did and did not bleach during the 2015 warming event (Fig 2B). This study was designed to assess the impacts of consecutive bleaching events on the early stages within the coral life cycle and selective processes already occurring in nature while also testing basic breeding techniques as an intervention strategy for coral restoration to maintain genetic diversity and promote resilience.

## Materials and methods

### 2.1 Selecting parent colonies and spawning events

*Montipora capitata* is a hermaphroditic broadcast spawner and its reproductive cycle, spawning dynamics, and early life stages have been extensively studied at the Hawaiʻi Institute of

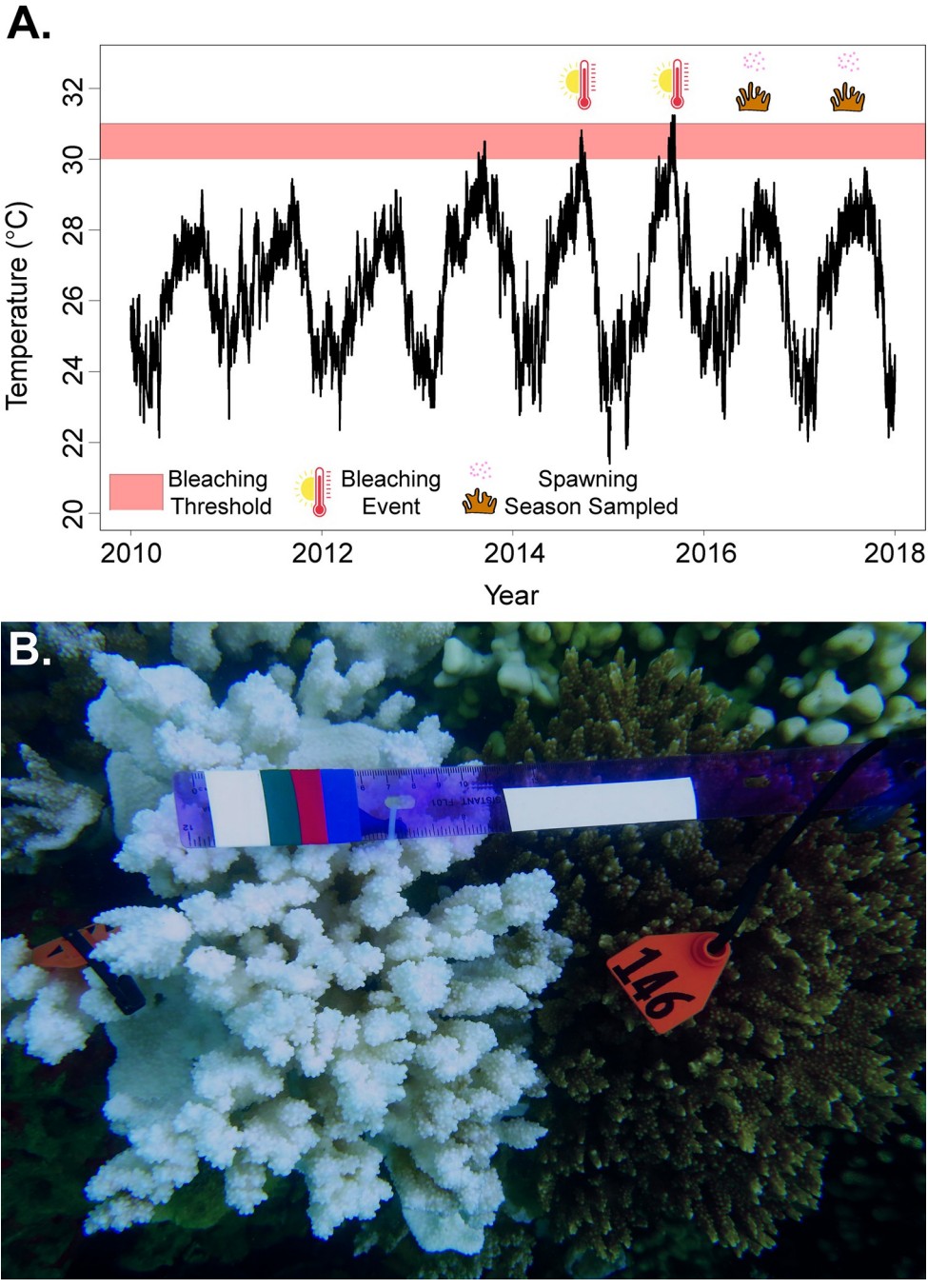

**Fig 1.** A) Temperature data from 2010 to 2017 (NOAA Buoy Moku o' Loe, HI Station ID: 1612480) illustrate historical patterns and identify years of bleaching events in Oʻahu, Hawaiʻi. The bleaching threshold between 30 to 31˚C of corals in Kāneʻohe Bay (Coles et al., 2018) is shown in the shaded red, thermometers indicate the 2014 and 2015 bleaching events and the spawning corals indicate the spawning seasons. B) An image depicting the tagged bleached (left) and nonbleached (right) parental colonies in response to the 2015 heat stress in Kāneʻohe Bay.

Marine Biology (HIMB) located in Kāneʻohe Bay, on the windward side of Oʻahu, Hawaiʻi, USA [27,60,61,63–65]. In Hawaiʻi, oogenesis begins a 9–10 month period as early as July and as late as October, while spermatogenesis begins the following April to May, ca. 1 month prior to the first spawning event in May or June [28], creating the potential for differential effects of

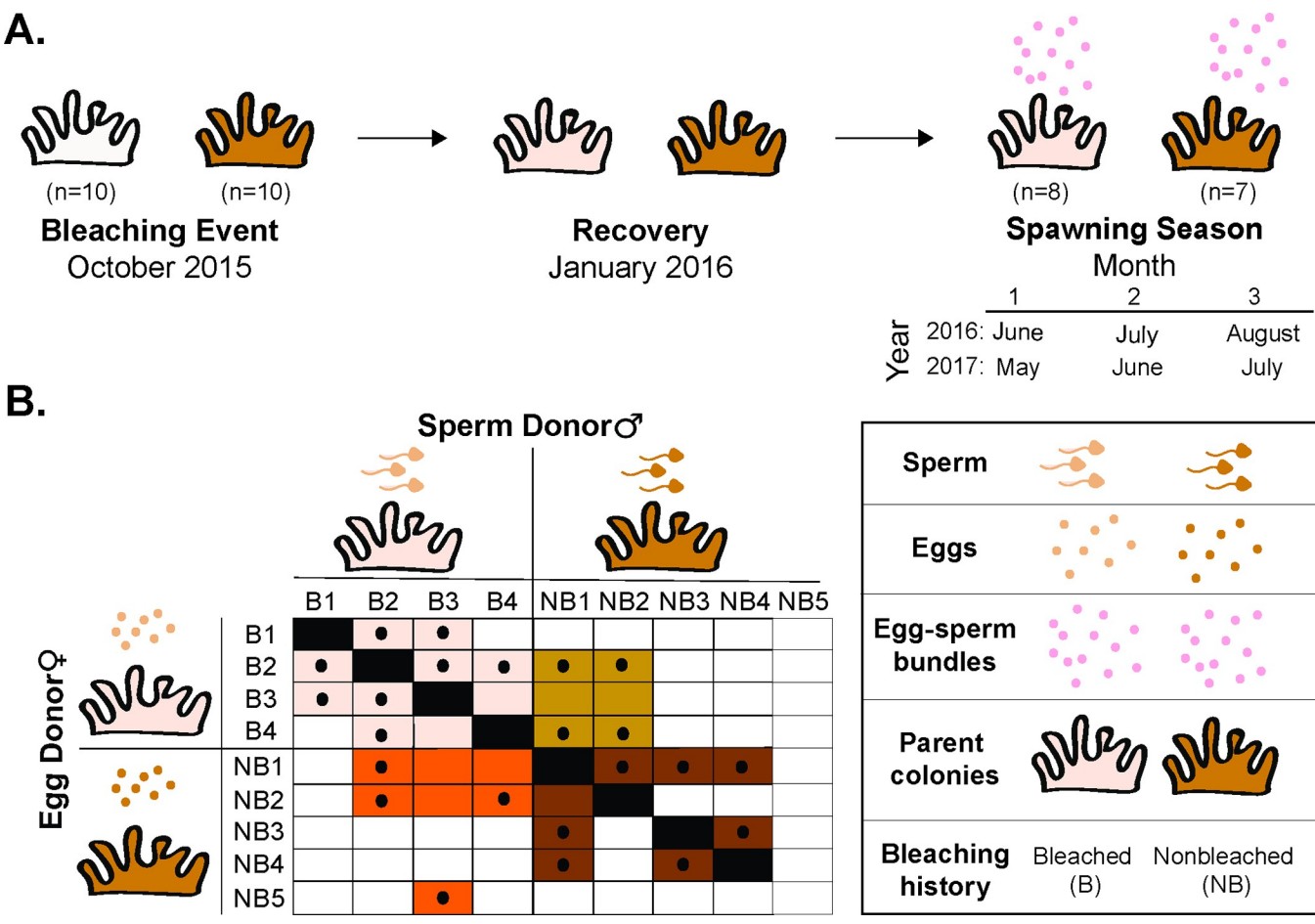

**Fig 2. Experimental design of the study.** A) Bleached and nonbleached colonies were tagged in October 2015 at the peak of the bleaching event. Bleached colonies in this experiment recovered by January 2016. Total reproductive output and gamete collections were measured during the 2016 and 2017 spawning seasons. Months of the spawning season differ between years because of the different timing of the new moon in 2016 and 2017. B) Selective breeding matrix illustrating the crossing of egg and sperm donors conducted in July 2016 based on parental bleaching history. Colored squares indicate the cross of individuals attempted and solid black circles indicate successful fertilization. Offspring from these crosses were used to measure survivorship of larvae and settlers and settlement.

bleaching on oocytes and sperm. Symbiodiniaceae are vertically transferred from *M. capitata* parent colonies into eggs prior to the formation of the egg-sperm bundles, which are released during spawning [63]. Spawning in *M. capitata* extends over three, consecutive lunar months between May and September for 3 to 5 consecutive nights between 20:45 and 22:30 hrs, starting on the night of the new moon [27,60]. The second and third nights are when the largest spawning events most commonly occur [60].

During the peak of the 2015 bleaching event in Hawai'i, ten pairs of colonies (30–100 cm diameter) of *M. capitata* were identified and tagged as bleached (B) and nonbleached (NB) along the leeward side of the reef surrounding HIMB (21'26.09 N, 157'47.47' W) on 20 October 2015 (Fig 3C). These colonies remained in the field until retrieved three days prior to the new moon of the spawning months in 2016 (June, July, and August) and 2017 (May, June, and July) (Fig 3A). To examine reproductive performance of B and NB colonies of *M. capitata*, parent colonies were collected by removing the entire colony from the reef, or by breaking large fragments (30–40 cm in diameter) from tagged colonies using a hammer and chisel. These collections were first completed on 4 and 5 June 2016. Of the twenty colonies tagged, seven

colonies that had not bleached and eight colonies that had bleached and recovered were alive and used for the study. The other five colonies not recovered had either died or were missing from the reef. The fifteen colonies were transported to the wet laboratory at HIMB in 20L buckets filled with seawater from Kāneʻohe Bay at an ambient temperature of ~28 to 29˚C. Colonies were randomly allocated to two ~1,300L shaded outdoor flow-through tanks [55,66]. Both tanks had sand-filtered seawater delivered at a flow rate of ~6L minute$^{-1}$ and a circulation pump (700 gph Magnetic Drive, Danner Manufacturing Inc. Islandia, NY, USA). Irradiance and temperature within each tank were recorded every fifteen minutes with a cosine corrected photosynthetically active radiation (PAR) sensor (Odyssey PAR loggers, Dataflow Systems Ltd, Christchurch, NZ) calibrated to a Licor 192SA sensor, and a temperature logger (Hobo™ Water Temp Pro v2 resolution ± 0.2˚C, Onset Computer Corporation, Bourne, MA, USA). Three to five days after each spawning event, colonies were returned to the original field site by attaching them to a fixed rack with cable ties and retrieved two days before the next new moon of the spawning season.

## 2.2 Sexual reproduction

Starting one night prior to the new moon, *M. capitata* parent colonies were monitored for seven nights. During each night of spawning, colonies were isolated at 19:30 in individual containers filled with ambient seawater from the flow-through tanks. When spawning occurred, *M. capitata* released egg-sperm bundles into the water column between 20:45 and 22:30 with peak spawning typically expected on the second night of the new moon [27,59,60]. Spawning activity of individual colonies was monitored each night and recorded as "spawn" or "no spawn". For the spawning colonies, we quantified the total volume of gametes released, number of eggs per bundle, and egg quality (i.e., area and abnormality).

Sterilized disposable pipets (2 mL) were used to gently collect all egg-sperm bundles at the water surface from each individual colony to avoid cross contamination or prematurely breaking the egg-sperm bundles. We preserved 3–5 egg-sperm bundles per colony per night to quantify the number of eggs per bundle, egg volume for size, and abnormality. Each egg-sperm bundle was placed in a 2 mL microcentrifuge tube and allowed to break up in 0.1 mL of seawater and for the eggs to hydrate for 2 hrs before preserved in zinc fixative (1:4 Z-fix, Sigma-Aldrich Inc. to 0.2 µm filtered seawater FSW). Preserved eggs from each bundle were photographed using an Olympus SZX7 dissecting microscope equipped with an Olympus America camera (SN: BH039933-H); from photographs, we counted the number of eggs per bundle and measured the egg diameter using ImageJ2 software (Schneider et al., 2012). Egg volume was calculated using the equation for a sphere with the measured egg diameter of spherical eggs. We also recorded the proportion of abnormal (irregular) eggs packaged within each bundle [36,63]. Remaining egg-sperm bundles from each colony were placed into individual 50 mL Falcon tubes to quantify the total volume of gametes of each colony per night. Annual reproductive output per colony was estimated by summing the spawn volume across the entire spawning season, normalized to planar surface area of the colony using Fiji software [67].

## 2.3 Fertilization success and Offspring

To compare offspring performance of bleached and nonbleached parents, we isolated the egg-sperm bundles from each parental colony that released more than 1 mL of spawn volume on the nights of 5 and 6 July 2016 (peak spawning) and placed egg-sperm bundles from each colony into a separate 50 mL falcon tube. Within one hour of the bundle breaking apart, eggs floated to the surface and sperm sank to the bottom. Sperm were pipetted from the bottom of

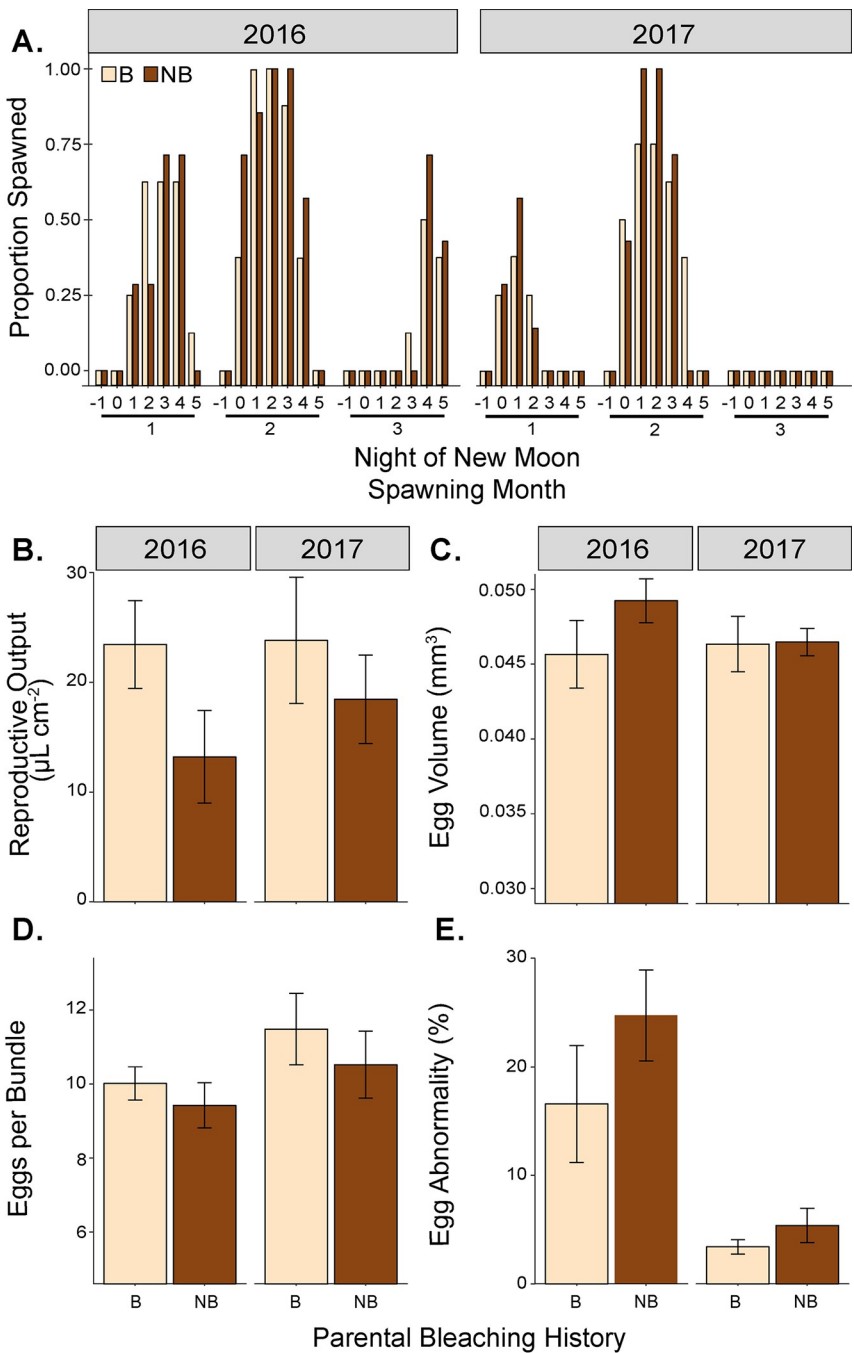

**Fig 3. Reproductive traits measured from the same parent colonies in Fig 2 during 2016 and 2017 spawning seasons following the 2015 bleaching event.** A) Proportion of spawning each night in 2016 and 2017 spawning seasons from parent colonies that had bleached and not bleached. "0" indicates the night of the new moon. Mean (± SE) values for B) reproductive output normalized to planar surface area, C) egg volume, D) number of eggs per bundle, and E) percent egg abnormality measured from bleached and nonbleached parents in 2016 and 2017.

the tube, and eggs were rinsed twice with 0.2 μm filtered seawater (FSW). Sperm from each colony was placed in separate 50 mL falcon tubes and later used to fertilize eggs from specific colonies. Nine colonies had adequate spawn volume to include in crosses, and thirty individual crosses were made from gametes based on parental bleaching history to generate four cross-

types (egg donor × sperm donor): B × B (n = 8), B × NB (n = 4), NB × B (n = 4), and NB × NB (n = 7) ([Fig 2B]). For fertilization, the eggs (1 mL) were in a concentration of ~$10^6$ sperm mL$^{-1}$ (by visual inspection) within a 50 mL falcon tube [68]. Thirty minutes after sperm and eggs were mixed, each cross type of fertilized eggs was transferred into individual 1 L conical tanks filled with UV-sterilized 1-μm FSW to avoid polyspermy. For *M. capitat*a, self-fertilization is extremely rare [68,69]. To estimate fertilization success, three subsamples of 20–30 eggs were collected from each conical after approximately 3-hrs (i.e., when initial cleave stages are expected [70,71]), placed in a 2-μL microcentrifuge tube, and preserved in Z-fix (1:4 Z-fix to FSW). Remaining embryos in the conical tanks developed, and slow flow rate of FSW was introduced to mitigate potential effects of montiporic acid [65]. Five days post-fertilization, 10–15 larvae per conical tank were placed in a 10 mL well-plate filled with 5 mL of FSW with a chip of crustose coralline algae to track settlement through time; FSW was exchanged every other day. The proportion of planulae and settlers were examined on days 7, 28, and 53 post-fertilization while the total number of offspring alive were counted on days 6, 7, 28, 53, and 59 post-fertilization to estimate survivorship probability curves.

## 2.4 Statistical analysis

All analyses were conducted in R (R Core Team, 2014; v. 3.5.1). We used a generalized linear mixed effects model to determine the effects of bleaching history on spawning activity, number of eggs per bundle, and egg abnormality of the 8 B and 7 NB parental colonies observed (*glmer* in *lme4*) [72] with a binomial (spawn/no spawn and proportion of abnormal eggs) and poisson (eggs per bundle count) response. Bleaching history (B/NB) and year (2016/2017) were included as fixed effects, and spawning month (1/2/3) and colony ID were included as random effects. To analyze total reproductive output and egg size, we used linear mixed effects models (*lme* in *lme4*) [72] with bleaching history and year as fixed effects, and colony ID as a random effect. Analysis of variance (ANOVA) tables were generated using type II sum of squares (*Anova* in *car*) [73]. Post-hoc analyses were conducted to further explore significant main effects and interactions. We utilized the emmeans package [74] to calculate and compare the estimated marginal means (EMMs), which represent the predicted means of the response variable for each level of the fixed effects, adjusted for the other covariates in the model. Pairwise comparisons between the levels of the fixed effects were then performed using Tukey's Honest Significant Difference (HSD) test to adjust for multiple comparisons. This approach allowed us to identify significant differences between specific treatment groups, while accounting for the variability associated with random effects.

To test the effects of parental bleaching history on offspring performance, we first analyzed the proportion of eggs fertilized using generalized linear mixed effects models with cross-type as a fixed effect and the egg donor and sperm donor as random effects. The proportion of eggs reaching each developmental stage (2-cell, 4-cell, 8-cell, and 16-cell), the Kruskal-Wallace test was applied as the dataset did not meet the assumption of normality. For post-hoc analysis, we performed the Dunn's test for multiple pairwise comparisons to determine which specific cross-types differed. To analyze the proportion of larvae that settled at 7, 28, and 59-days post-fertilization, we used a generalized linear mixed effects model with cross-type and day (7, 28, and 59-d post-fertilization) as fixed effects and colony ID of egg donor and sperm donor as random effects. Lastly, we generated survivorship estimate curves to visualize offspring fate by cross-type with *ggsurvplot* of the census over time (i.e., days 6, 7, 23, 27, 28, 53, and 59 post-fertilization) (*survfit* in *survminer*) [75]. Cox proportional hazards (CPH) model was used to analyze the effects of cross, egg donor, and sperm donor individually on offspring survivorship (*coxph* in *survminer*) [75]. Dispersion parameters were inspected through a simulation-based approach (*DHARMa* package) [76].

## Results

### 3.1 Sexual reproduction and egg traits

All fifteen colonies observed in this study released egg-sperm bundles one or more nights in both years (Fig 3A). When spawning was observed, colonies began releasing egg-sperm bundles between 20:20 and 21:32 hrs and ended between 20:30 and 22:15 hrs. Parental bleaching history did not affect the occurrence of spawning ($P = 0.619$) and had no interactive effect with year ($P = 0.982$). The proportion of colonies releasing gametes significantly differed by year ($P < 0.001$) which may largely be due to some spawning in 2016 compared to no spawning in 2017 during the third month (August). In 2017, the proportion of colonies participating in spawning events was 36% lower than in 2016. In both years, the second month of the spawning season had the highest proportion of colonies spawning.

In 2016, the spawning season following consecutive bleaching events, colonies that bleached and recovered had 22.5% higher mean total reproductive output than colonies that did not bleach, although this was not statistically significant (Fig 3B; Table 1; P = 0.076). There was no effect of year and no interaction between bleaching history and year on reproductive output (Fig 3B; Table 1; $P \geq 0.560$). Individual egg volume ranged from 0.032 to 0.099 mm³ and did not differ by parental bleaching history, year, or by their interaction (Fig 3C; Table 1; $P \geq 0.462$). The number of eggs per bundle from both bleached and nonbleached parental colonies ranged from 2 to 29, and mean eggs per bundle for all colonies examined was 13.3% less in 2016 than in 2017 (Fig 3D; Table 1; P = 0.017). Eggs per bundle did not differ by parental bleaching history (Fig 3D; P = 0.249). There were 79.5% more eggs with irregularities in 2016 than in 2017 (P < 0.001) with no difference by bleaching history (Fig 3E; Table 1; P = 0.292).

### 3.2 Fertilization, survivorship, and settlement

While reproduction continued in the colonies examined, we found that cross-type did have an effect on fertilization, embryonic development, and percent larval survivorship (Fig 4;

**Table 1. Statistical summary of Type II Wald χ² test of generalized linear mixed effects model and linear mixed effect models testing the fixed effects of spawning year and parent history of bleaching susceptibility on sexual reproduction.**

| Response Variables | Fixed Effects | $\chi^2$ | df | *P*-value | Post-hoc Summary |
|---|---|---|---|---|---|
| Colony-level Spawning | Bleaching History | 0.248 | 1 | 0.619 | |
| (0 = no spawn / 1 = spawn) | **Year** | 22.479 | 1 | **< 0.001** | **2016 > 2017** |
| | Bleaching History * Year | 0.001 | 1 | 0.982 | |
| Total Reproductive Output Log transformed | Bleaching History | 3.155 | 1 | 0.076 | |
| | Year | 0.339 | 1 | 0.560 | |
| | Bleaching History * Year | 0.097 | 1 | 0.756 | |
| Egg Volume | Bleaching History | 0.108 | 1 | 0.742 | |
| | Year | 0.541 | 1 | 0.462 | |
| | Bleaching History * Year | 0.225 | 1 | 0.635 | |
| Eggs per Bundle | Bleaching History | 1.332 | 1 | 0.249 | |
| | **Year** | **5.656** | 1 | **0.017** | **2016 < 2017** |
| | Bleaching History * Year | 1.408 | 1 | 0.235 | |
| Egg Abnormality | Bleaching History | 1.109 | 1 | 0.292 | |
| | **Year** | **191.259** | 1 | **<0.001** | **2016 > 2017** |
| | Bleaching History * Year | 0.035 | 1 | 0.852 | |

Significance indicated in bold text.

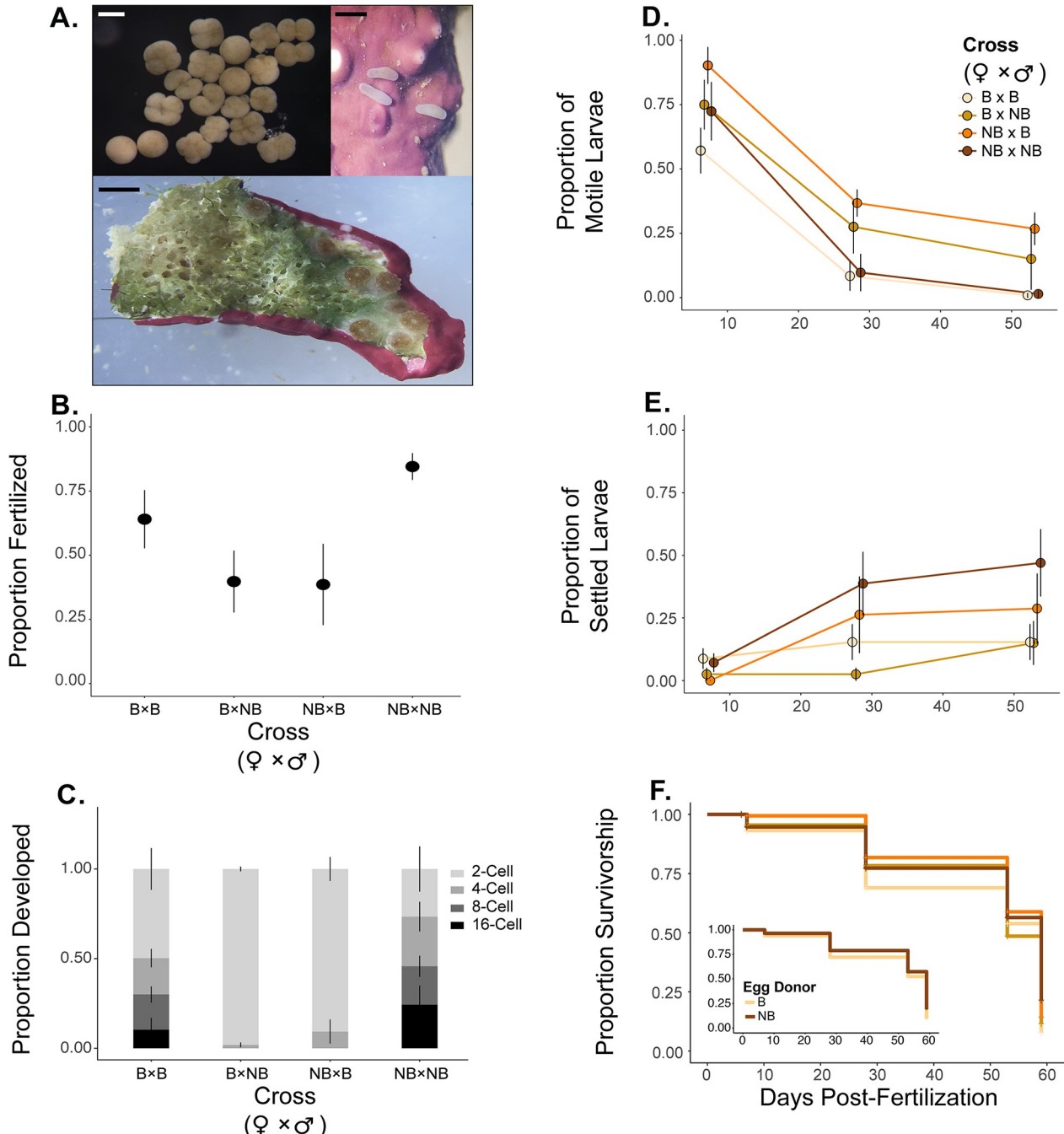

**Fig 4. Offspring performance from selected crosses.** A) Images of fertilized eggs and embryos (scale bar = 500 μm), planula larvae (scale bar = 500 μm), and settlement (1 mm). Mean ± SE. B) proportion of eggs fertilized by cross-type, C) proportion of cell division after 3-h fertilization D) proportion of motile larvae and E) settlers during five timepoints over a 59-d period, and F) survivorship estimate curves by cross over seven timepoints between 6 and 59-d with the figure embedded comparing the survivorship curves of offspring from bleached and nonbleached egg donors.

Table 2). Specifically, fertilization success in the NB × NB cross-type was higher than the B × NB and NB × B cross-types (Fig 4B; Table 2; post-hoc P = 0.002 and 0.010, respectively) but not B × B (post-hoc P = 0.163). The fertilization success in cross-type B × B also did not differ between NB × B (post-hoc P = 0.250), but was higher than B × NB (post-hoc P = 0.047).

**Table 2. Statistical summary of Type II Wald $\chi^2$ test of generalized linear mixed effects model testing the fixed effects of cross-type (NB × NB, NB × B, B × NB, B × B) on the proportion of fertilized embryos and summary of the Kruskal-Wallace test for the cellular development (2-Cell, 4-Cell, 8-Cell, and 16-Cell).** Statistical summary of Type II Wald $\chi^2$ test of generalized linear mixed effects model testing the fixed effects of cross-type on larval survivorship, and settlement over three timepoints post-fertilization.

| Response Variable | | Effect | $\chi^2$ | df | P-value | Post-hoc Summary |
|---|---|---|---|---|---|---|
| Embryonic Development | Fertilization | **Cross** | **16.334** | **3** | **0.001** | B×B > B×NB<br>NB×NB > B×NB, NB×B |
| | 2-Cell | | **34.071** | **3** | **<0.001** | B×B = NB×NB<br>NB×B = B×NB |
| | 4-Cell | | **21.729** | **3** | **<0.001** | B×B = NB×NB, NB×B<br>NB×B = B×NB |
| | 8-Cell | | **33.882** | **3** | **<0.001** | B×B = NB×NB<br>B×NB = NB×B |
| | 16-Cell | | **20.445** | **3** | **<0.001** | B×B = NB×NB, B×NB, NB×B<br>NB×B = B×NB |
| **Response Variable** | **Fixed Effects** | | | | | |
| Larval Survival<br>*Square-root transformed* | **Cross** | | **20.915** | **3** | **<0.001** | B×B ≠ NB×B |
| | **Days Post-Fertilization** | | **178.595** | **2** | **<0.001** | |
| | Cross * Days Post-Fertilization | | 7.174 | 6 | 0.305 | |
| Larval Settlement | Cross | | 7.623 | 3 | 0.055 | |
| | **Days Post-Fertilization** | | **17.214** | **2** | **<0.001** | |
| | Cross * Days Post-Fertilization | | 9.066 | 6 | 0.170 | |

Significance indicated in bold text.

Cell division advanced beyond the 2-cell stage more quickly for within cross-types (B × B and NB × NB) than between cross-types (B × NB and NB × B) at 3-h post-fertilization. Embryos from both B × B and NB × NB cross-types reached the 16-cell stage at 3-h post fertilization, whereas embryos from B×NB and NB × B crosses developed at a slower rate and only reached the 4-cell stage (Fig 4C, Table 2).

Percent larval survivorship and settlement varied by cross-type, driven by egg donor bleaching history (Fig 4D and 4E; Table 3). Offspring developed from eggs from previously B egg donors had lower survivorship than those from NB egg donors. NB egg donors had a significant effect on the proportion of larvae survival (Fig 4E; $P < 0.001$). However, no difference was found in offspring survivorship from bleached or nonbleached sperm donors ($P = 0.992$). Overall, percent mortality from the initial to final time point (i.e., day 5 to 59) were 92.5% for B × B, 87.8% for B × NB, 85.6% for NB × B, and 77.3% for NB × NB (Fig 4F).

**Table 3. Summary of Cox proportional hazards analysis of coral offspring survival influenced by the fixed effects: Cross-type, dam, and sire over time with model average estimates of the hazard ratio (with 95% confidence intervals; Cross (NB × NB, NB × B, B × NB, B × B): df = 3 or egg/sperm donor (NB vs. B): df = 1; $n = 1,318$; number of events = 560) for five timepoints (day 6, 7, 28, 53, and 59).**

| Fixed effect | Hazard ratio | z | P-value |
|---|---|---|---|
| **Cross** | 0.90 (0.84–0.96) | -3.071 | **0.002** |
| **Egg Donor** | 0.77 (0.65–0.91) | -3.068 | **0.002** |
| Sperm Donor | 1.00 (0.80–1.24) | -0.010 | 0.992 |

Significance indicated in bold text.

## Discussion

Here, we demonstrate the influences of marine heatwaves on coral reproductive capacity and parental effects in spawning seasons following major bleaching events. It is noteworthy that the unprecedented, consecutive warming events in 2014 and 2015 in Kāneʻohe Bay, Hawaiʻi influenced the reproductive capacity of *M. capitata* regardless of *parental bleaching response*. When comparing the first spawning season following the 2015 bleaching event to the second, *M. capitata* colonies had fewer eggs packaged within the egg-sperm bundles released. While average egg volume did not differ between years, the egg abnormality was higher in 2016 than in 2017 regardless of parental bleaching history. Parental colonies that bleached and did not bleach had similar reproductive output, number of eggs per bundle and egg abnormality. However, delayed beneficial maternal effects were observed in offspring from parents resistant to bleaching. These results demonstrate that although *M. capitata* has the energetic capacity to continue reproduction despite bleaching response, cross-generational impacts occur (Byrne et al., 2020), with possible ecological consequences downstream.

### 4.1 Reproductive capacity after bleaching events

*M. capitata* appears to maintain reproductive resilience, as well as recovery with time, after consecutive marine heatwaves and coral bleaching events, as evidenced by continuing synchronous broadcast spawning and production of viable eggs and sperm. These results are consistent with prior studies examining the influence of environmental and biological factors on *M. capitata* gametogenesis and spawning in Kāneʻohe Bay [28,59]. For instance, Padilla-Gamiño et al. [28] found similar rates of gametogenesis along a strong sedimentation gradient. Further, Cox [59] found no differences in reproductive output, eggs per bundle, and egg size between B and NB parents in the spawning season immediately following the 2004 mild warming event. Resilience in *M. capitata* may be due to its capacity to maintain energetic stability under stress [53], here evident by the completion of gametogenesis even at the cost of producing fewer eggs per bundle with higher proportion of irregularity in shape in 2016 than in 2017. One hypothesis to explain similar reproductive traits in bleached and nonbleached parents, is that after the thermal stress (Septemebr-October), there is still time for the colonies to recover (~5–6 months) and develop gametes that can be released during the spawning season (May-August) [14,53,60,63]. Furthermore, Rodrigues & Padilla-Gamino [77] found that *M. capitata* colonies that bleached allocated 10% more carbon to gametes despite bleaching by limiting the allocation of carbon to adult tissues, with 50–80% less carbon allocated to bleached compared to non-bleached colonies. Compared to other species, *M. capitata* prioritizes gametogenesis at the expense of the adult colony. Maintaining egg traits such as size and biochemical composition would serve as an advantageous strategy to ensure ecological fitness of parents and their developing offspring [61,78,79]. For example, there may be an optimal egg size that needs to be achieved to ensure successful fertilization [80,81]. It is notable that the relationship between egg size and number of eggs per bundle in our study has shifted from prior studies; we found 10–12 eggs per bundle in 2016–2017 compared to 15–18 eggs per bundle in studies and egg size was 11% larger in our study than previous studies [59,60]. This apparent tradeoff in reproductive effort suggests plasticity in response to environmental changes and emphasizes the need for long-term studies to detect changes in sexual reproduction [14,35,36]. While further examination of egg traits, such as total lipid content and composition of lipid classes, was beyond the scope of this study due to limited material available, larger egg volume could be beneficial in storing lipids and carbohydrates as well as increased surface area for slower sperm to fertilize eggs in the water column.

High inter- and intraspecific variation in thermal tolerance contribute to reproductive consequences after bleaching events [8,17,33]. For example, there were no differences in percent reproductive polyps between bleached and nonbleached colonies of acroporid species at Heron Island on the Great Barrier Reef after the 1998 bleaching event [82]. Baird and Marshall [8] found that the bleaching response of *Acropora millepora* did not influence fecundity, whereas the bleaching response of *Acropora hyacinthus* strongly influenced the completion of gametogenesis. It is important to emphasize that although reproductive capacity after bleaching events can be greatly suppressed, there are species and populations that are resistant and/or more able to recover from bleaching [7,8,17,39,59,82]. Distinctive populations carrying resilient individuals are critical to identify and protect, particularly if they are successful in continuing sexual reproduction to replenish impacted neighboring reefs [83,84]. Coral reproductive modes and strategies have evolved to withstand environmental fluctuations and severe selective pressures, but the question of how much thinning can a population withstand without complete collapse remains.

## 4.2 Parental effects on fertilization and offspring survivorship

We demonstrate parental effects, or cross-generational plasticity, in *M. capitata*, with parent cross-type having an effect on fertilization and embryonic development with maternal effects apparent in offspring survivorship. Fertilization success differed by cross-type which may be due to gametic compatibility [85]. Such compatibility could be driven by gamete-recognition proteins that mediate fertilization through chemoattraction, binding, and fusion of egg and sperm [85–87]. Furthermore, high gamete compatibility may explain the advanced rate in cell division during embryogenesis in offspring from NB × NB and B × B cross-types. The lack of compatibility observed in crosses between B × NB and B × NB could potentially result from lineage crossing. However, in our study, we were unable to analyze the genetic composition of the parent organisms and we could not determine if they belonged to distinct parental lineages. Future studies should take into account parental lineages to better understand gamete compatibility, inheritance patterns and traits that can lead to increased genetic diversity or novel offspring phenotypes.

Egg-sperm compatibility has been observed as a mechanism for pre-zygotic isolation to select for populations that are likely to succeed under intense environmental pressures, such as temperature [88–91]. With regards to sperm selection, Henley et al. [92] demonstrated sperm motility in *M. capitata* is strained with a severe decline that may be associated with damaged mitochondria in response to heat stress. Eggs from parent colonies that were resistant to bleaching had offspring with notably higher survivorship regardless of the sperm donor bleaching history [42]. More pronounced benefits of nonbleached egg donors support previous work of maternal provisioning in coral offspring [93–95]. Previous studies have demonstrated that beneficial cross-generational plasticity can occur from maternal effects observed in offspring survivorship. Benefits of maternal effects could be associated with phenotypic traits that help overcome hurdles created by thermal stress such as energetic provisioning through lipid reserves stored in the eggs and larvae [61,96,97], mitochondria [96], or vertical transmission of Symbiodinaceae from the parent into the eggs [64,94,97].

*M. capitata* houses the endosymbionts *Cladocopium* spp. and *Durusdinium* spp., formerly Clade C and D, respectively. It has been shown that *M. capitata* colonies associate with *Durusdinium* spp. in more challenging environments such as high light and variable thermal regimes [61,98,99]. After a bleaching event, there was a rise in the relative proportion of the heat-tolerant symbiont *Durusdinium* spp. in *M. capitata* colonies across most areas in Kāneʻohe Bay [98]. However, despite this increase, the overall composition of Symbiodiniaceae symbionts

remained largely unchanged, and distinct regions of the bay retained their pre-bleaching compositions. In *M. capitata*, these symbionts are vertically transferred to the eggs creating offspring with different assemblages [64] that could confer different physiological attributes to the offspring. For example, Little et al. [100] found that *Acropora* juveniles grew faster when infected with *Cladocopium* spp. than *Durusdinium* spp. (formerly clade C and D, respectively) and Abrego et al. (2008) showed enhanced physiological tolerance and higher [14]C photosynthate incorporation in juveniles infected with *Cladocopium* spp. (clade C1). Padilla-Gamiño et al. [64] showed that *Cladocopium* spp. is more likely to be transferred to *M. capitata* eggs, but further research is needed to better understand transfer mechanisms, and how different symbionts influence survival, tolerance and/or tradeoffs in larvae and juveniles.

### 4.3 Interventions for thermal tolerance

Research on coral reefs has become greatly focused on identifying human interventions (i.e., assisted evolution) that support biological persistence and resilience against anthropogenic stressors [101–103]. Developing effective interventions to implement has become increasingly urgent to protect shallow-dwelling coral reef ecosystems [103]. Current strategies proposed to overcome bottlenecks in early life history include identifying genetic adaptation [104], environmental hardening through non-genetic or epigenetic mechanisms [105–110], manipulation of Symbiodiniaceae symbionts [41,111,112], cryopreservation for coral conservation [113], and selective breeding [94,95,114].

Human interventions, such as selective breeding in coral sexual propagation, has been proposed as one of the viable options to maintain genetic diversity and increase resilience in restoration efforts [15,102,103,105,115,116]; however, feasibility to potentially scale up efforts remain limited and costly without full understanding of tradeoffs [117,118]. Our study supports the potential for selective breeding and environmental hardening to have positive fitness consequences. In our study, bleaching in *M. capitata* did not severely disrupt reproductive output or egg traits measured (size and abnormality), but the use of eggs from NB colonies in the intentional crossing of gametes produced offspring with higher settlement and survivorship, while bleached corals had higher overall fecundity to balance reduced survivorship and settlement. These results are important to maximize restoration efforts through selective breeding by identifying candidate colonies in the natural environment or through manipulated stress tests and performing crosses using the gametes of resilient colonies. We encourage further research to test the efficacy and trade-offs of human-assisted evolution, particularly selective propagation and environmental hardening, designed to increase coral resistance that would ensure the continuation of coral reefs confronted by global climate change.

### Acknowledgments

We would like to thank members of the Gates Coral Laboratory for their technical support and advice, especially Jen Davidson and Dr. James Guest. We are grateful to Mary Hagedorn, Amy Moran, and Peter Marko for their feedback on the manuscript, the many volunteers especially Dyson Chee, Megan Buras, Katie Allen, Shayne Fabian, and Kat McPherson and the security Greg Miranda and Moses at Moku o Loʻe who ensured safety and the success of this research. We dedicate this research to Dr. Ruth D. Gates and her infectious enthusiasm that pushed assisted evolution to the forefront of coral biology–you rock!

### Author Contributions

**Conceptualization:** Elizabeth A. Lenz, Megan J. Donahue, Ruth D. Gates, Hollie M. Putnam, Jacqueline L. Padilla-Gamiño.

**Data curation:** Elizabeth A. Lenz, Eveline van der Steeg.

**Formal analysis:** Elizabeth A. Lenz, Megan J. Donahue, Jacqueline L. Padilla-Gamiño.

**Funding acquisition:** Elizabeth A. Lenz, Ruth D. Gates, Hollie M. Putnam.

**Investigation:** Elizabeth A. Lenz.

**Methodology:** Elizabeth A. Lenz, Ruth D. Gates, Hollie M. Putnam, Eveline van der Steeg.

**Project administration:** Elizabeth A. Lenz, Eveline van der Steeg.

**Resources:** Elizabeth A. Lenz, Ruth D. Gates.

**Software:** Elizabeth A. Lenz.

**Supervision:** Megan J. Donahue, Ruth D. Gates.

**Validation:** Elizabeth A. Lenz.

**Visualization:** Elizabeth A. Lenz.

**Writing – original draft:** Elizabeth A. Lenz, Jacqueline L. Padilla-Gamiño.

**Writing – review & editing:** Elizabeth A. Lenz, Megan J. Donahue, Hollie M. Putnam, Eveline van der Steeg, Jacqueline L. Padilla-Gamiño.

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
