## [Decision Letter · Decision Letter 0]

5 Nov 2023

PONE-D-23-24872Parental effects provide an opportunity for coral resilience following major bleaching eventsPLOS ONE

Dear Dr. Lenz,

Thank you for submitting your manuscript to PLOS ONE. After careful consideration, we feel that it has merit but does not fully meet PLOS ONE’s publication criteria as it currently stands. Therefore, we invite you to submit a revised version of the manuscript that addresses the points raised during the review process.

We look forward to receiving your revised manuscript.

Kind regards,

Atsushi Fujimura

Academic Editor

PLOS ONE

“National Science Foundation Graduate Research Fellowship Program

Paul G. Allen Family Foundation.”

“This work was supported by funding from Paul G. Allen Family Foundation to RG and the National Science Foundation Graduate Research Fellowships to EAL. We would like to thank members of the Gates Coral Laboratory for their technical support and advice, especially Jen Davidson and Dr. James Guest, we are grateful to Mary Hagedorn, Amy Moran, and Peter Marko for their feedback on the manuscript, the many volunteers especially Dyson Chee, Megan Buras, Katie Allen, Shayne Fabian, and Kat McPherson and the security Greg Miranda and Moses at Moku o Loʻe who ensured safety and the success of this research. We dedicate this research to Dr. Ruth D. Gates and her infectious enthusiasm that pushed human-assisted evolution to the forefront of coral biology – you rock!”

“National Science Foundation Graduate Research Fellowship Program

Paul G. Allen Family Foundation.”

6. We note that Figures 1B, 2, and 4A in your submission contain copyrighted images. All PLOS content is published under the Creative Commons Attribution License (CC BY 4.0), which means that the manuscript, images, and Supporting Information files will be freely available online, and any third party is permitted to access, download, copy, distribute, and use these materials in any way, even commercially, with proper attribution. For more information, see our copyright guidelines: http://journals.plos.org/plosone/s/licenses-and-copyright.

1. You may seek permission from the original copyright holder of Figures 1B, 2, and 4A to publish the content specifically under the CC BY 4.0 license.

7. Please include your tables as part of your main manuscript and remove the individual files. Please note that supplementary tables should be uploaded as separate "supporting information" files.

Additional Editor Comments:

Message from the Editorial Office regarding a deceased co-author: For a deceased co-author, this would need to be indicated in the title page of the manuscript in addition to the acknowledgments. When submitting your manuscript, as the corresponding author you must accept the responsibility for the integrity of all data collected and analyzed by the deceased author. Suggested text for this in the Acknowledgements section is "[Author] passed away before the submission of the final version of this manuscript. [Corresponding author] accepts responsibility for the integrity and validity of the data collected and analyzed." Please also add a "†" symbol next to the author's name in the author list and include a note that this author is deceased.

Reviewers' comments:

Reviewer's Responses to Questions

**Comments to the Author**

1. Is the manuscript technically sound, and do the data support the conclusions?

Reviewer #1: Partly

Reviewer #2: Yes

Reviewer #3: Partly

2. Has the statistical analysis been performed appropriately and rigorously? 

Reviewer #1: Yes

Reviewer #2: Yes

Reviewer #3: Yes

3. Have the authors made all data underlying the findings in their manuscript fully available?

Reviewer #1: Yes

Reviewer #2: Yes

Reviewer #3: Yes

4. Is the manuscript presented in an intelligible fashion and written in standard English?

Reviewer #1: Yes

Reviewer #2: Yes

Reviewer #3: Yes

5. Review Comments to the Author

Reviewer #1: Reviewer comments on lines following "~".

Direct quotes being addressed from manuscript are in "".

Review comments are broken up by sections and sub-headers in sections.

Introduction:

“Coral bleaching is known to impact sexual reproduction (Baird & Marshall, 2002; Fisch et al.,2019) and recruitment (Hughes et al. 2019; Price et al., 2019)… colonies that bleached and recovered (Ward et al., 2002).”

~ This reads like regurgitated info. I recommend adding a concluding sentence to this paragraph that highlights what you want the reader to understand about the importance of your project from the inclusion of this paragraph in the intro.

~ There is differential bleaching observed dramatically across individuals so close to each other on the same reef. Has microsatellite work been done on these colonies? Is there a possibility of cryptic species/lineages across the bleaching susceptible and resistant corals here?

~ The authors talk a lot in the intro about how bleached individuals do not produce as many eggs as individuals that did not bleach for several coral species, but the authors findings (and included previous work on M. capitata) do not support this. Why include so many examples of this information in the intro then? Maybe just say concisely in discussion.

“This reproductive response may contribute to its ecological success along the fringing and patch reefs of Kāneʻohe Bay (Kolinski, 2004).”

~ This statement is contradictory to some of the authors’ explanations of their results and statements in the discussion. I would consider including this association with different wording. Maybe say “This reproductive response may have previously contributed..” since you are talking about historical characteristics that may have been beneficial to the reproductive efforts of this species, but the authors are now highlighting how these characteristics may no longer be beneficial for some individuals in this paper.

“This study was designed to assess selective processes in nature confronted by climate change while also exploring breeding techniques as an intervention strategy for coral restoration to maintain genetic diversity.”

~ I like this statement that tries to summarize the importance of this study at the end of the intro, but the wording is a bit confusing. Rework this sentence for more clarity.

Methods:

2.1 Selecting parent colonies and spawning events

~ The first paragraph of this section reads like it should be included in the introduction, instead of in the methods. Also, you highlight that Symbiodiniaceae symbionts are provided by the mothers to the eggs. Does this study assess dominant symbiont association of the mothers and/or larvae produced? That would help a lot with connecting impacts observed to more and less beneficial symbiont associations for this species.

2.3 Fertilization success and larval survivorship

“Five days post-fertilization, 10-15 larvae per conical tank were placed in a 10-mL well-plate filled with 5-mL of FSW”

~ Hyphens should be removed between 10 and mL, along with 5 and mL.

Results:

3.1 Sexual reproduction and egg traits

“The proportion of colonies releasing gametes significantly differs by year (P < 0.001). In 2017, the proportion of colonies participating in spawning events was 36% lower than in 2016.”

~ It would help the reader to have the actual number of colonies that spawned for each group and each year listed here (or maybe in a table). Can also be in parentheses stated in a sentence.

~Also, this looks like some legacy effect of heat stress across B and NB since both had less spawning individuals in 2017. Would the findings for egg production and frequency of abnormal eggs not also represent possible legacy effects from the bleaching events? Might be an important consideration for how long it actually takes corals to fully “recover” from bleaching, whether or not they were visibly bleached.

“In 2016, the spawning season following consecutive bleaching events, colonies that bleached and recovered had 22.5% higher mean total reproductive output than colonies that did not bleach, although this was not statistically significant”

~ What explanation do the authors have for this since this seems contrary to bleached individuals being less fit than individuals that did not bleach.

~ Was any data collected to make sure the bleached and unbleached colonies were not in fact cryptic lineages/species? If not, then the incompatibility in BxNB and NBxB crosses could be a product of lineage crossing rather than the listed reasons by the authors. Indicating the authors might not be comparing apples to apples. This option should also be included in the manuscript to explain the differences in results if there was no testing of these colonies to eliminate this possibility.

3.2 Fertilization, survivorship, and settlement

“There were no differences between BxB and the other three cross-types (P ≥ 0.342).”

~ the font changes here, make sure this is formatted correctly

Cell division was faster for within than between type crosses.

~ Cryptic lineages could explain this as well.

Egg donor’s bleaching history affects Percent swimming larvae and settlement of larvae. Along with lower survivorship of larvae from bleached donors. History of parent did not affect sperm.

BxB had highest mortality(~92%), then BxNB, and finally NBxNB (~77%) over 59 days.

~Differences in egg quality? lipid class analyses of the eggs from B and NB would greatly benefit the impact and explanation of findings in this paper, due to the observed impacts of B and NB eggs on larvae survival. This would allow comparison of egg components, which might be driving differences in larval survival since eggs were of comparable volumes across B and NB groups and the results could be compared to previous measures of egg lipids from cited papers within this manuscript. If samples are available, the authors might consider adding these results.

Discussion:

“These results demonstrate the impacts of marine heatwaves and coral bleaching on reproductive capacity in the following spawning season with the capacity to recover in subsequent nonbleaching years.”

~ I agree with this statement by the authors, but I think it should also be emphasized here that similar impacts to egg production were observed whether or not the colonies were visibly bleached (B or NB) in this study. This supports that corals that are not bleached are also suffering some of the same impacts (i.e., the number of spawning individuals, number of eggs produced per bundle, number of abnormal eggs, etc.) as bleached corals after bleaching events, even if they are better positioned to survive and recover than bleached individuals.

4.1 Reproductive capacity after bleaching events:

~ Authors should define what is meant by “recover” here. Colonies after bleaching events are more susceptible to disease for more than a year after and can have reduced energy reserves for longer than that. If you consider these part of “recovery” of the coral, then ~5-6 months is not long enough to consider the corals as recovered. The authors also note in the intro that “The impacts of coral bleaching may last for months to years after the initial thermal stress” and cite a paper indicating 3 to 5 years for coral recovery, both of these statements contradict the statement made in the discussion of ~3 to 5 months for recovery. The authors need to be consistent in how they define “recovery” throughout the manuscript.

“Maintaining egg traits such as size and biochemical composition would serve as an advantageous strategy... long-term studies to detect changes in sexual reproduction (Levitan et al., 2014; Hagedorn et al., 2016; Price et al., 2019; Schlesinger et al., 2019).”

~ Why would eggs be larger than before? Authors should explain their thoughts on this finding more.

~ Also, this might be a good place to highlight that the plasticity utilized by B and NB may have been equally beneficial for B and NB before but that now egg quality might be suffering for B mothers due to continued impacts.

4.2 Parental effects on fertilization and offspring survivorship:

“In this study, bleached colonies had similar fertilization success to nonbleached colonies, the thermally tolerant individuals will likely have a stronger selective advantage as warming continues to weed out thermally sensitive adults and their offspring (Drury et al., 2022).”

~ I do not understand this statement, authors need to clarify more here. How can the B and NB colonies having similar fertilization success mean there is stronger selective advantage for NB adults and offspring? I understand why the authors are trying to link in natural selection towards more resilient individuals, but that is contrary to the results discussed in the beginning of this sentence. Especially since similar fertilization, number of eggs, and volume of eggs were found across both these groups for both years following the bleaching events. Rework this statement for clarity.

“Beneficial cross-generational plasticity through maternal effects observed in offspring survivorship may be attributed to energetic provisioning through lipid reserves stored in the eggs and larvae (Jones et al., 2011; Padilla-Gamiño et al., 2013; Rivest et al., 2017), mitochondria (Dixon 15 et al., 2015), or vertical transmission of Symbiodinaceae from the parent into the eggs (Jones et al., 2010;Padilla-Gamiño et al., 2012; Quigley et al., 2016).”

~ This statement should be fleshed out more and clarified further rather than just being a list of semi-unrelated traits. For example, what is meant by “Mitochondria” in this list because it does not look like it is connected to energy reserves.

“M. capitata houses the endosymbionts Cladocopium spp. and Durusdinium spp., formerly Clade C and D, respectively.”

~ It is outdated and improper to list genera levels of specific symbiont-host associations. Since only one coral species is assessed in the current study the specific symbiont species that associate with this coral in this area should be listed instead (i.e., Durusdinium trenchii instead of “Durusdinium spp.”)

~ Also, Padilla-Gamiño et al., 2013 does not actually identify the symbionts in M. capitata so it would be more correct to cite Padilla-Gamiño et al. 2012 in PLoSOne instead, or better yet de Souza et al. 2022 since that has more recent and specific symbiont identifications for M. capitata in Kāne‘ohe Bay.

“In M. capitata, these symbionts are vertically transferred to the eggs creating offspring with different assemblages (Padilla-Gamiño et al., 2013) that could confer different physiological attributes to the offspring.”

~ This should also be Padilla-Gamiño et al., 2012 PLoSOne that looked at symbionts in eggs, rather than 2013.

“For example, Little et al. (2004) found that Acropora juveniles grew faster when infected with clade C than clade D, and Abrego et al. (2008)”

~ Same comment about symbiont names here. Look up the specific genera for these symbiont-host associations. It is not appropriate to call the symbionts by the antiquated “Clades” designations, also the authors should be consistent with how they reference the symbionts throughout the manuscript rather than changing between genera names and clade names.

“Our results suggest that nonbleached colonies have higher ecological fitness than their bleached counterparts and may benefit the maintenance of genetic diversity while guiding populations towards higher thermal tolerance in a warming ocean.”

~ Authors discuss the importance of the photosynthetic endosymbiont to the host and that this symbiont is vertically transmitted to the larvae by the mother. After disturbance the dominant Symbiodiniaceae community members can shift rapidly back to a previous non-disturbed community state (sometimes within a few months to a year, see references below this comment). It is important to know what symbionts were at play in B and NB parents (and the sampled larvae) before making any statements about what impacts these may have had on the larvae survivorship in the current study.

Were the symbionts ever identified for the parent colonies or subset of larvae to determine if any claims in this paragraph were relevant to the corals and findings of this study? If not, then the concluding sentence of this paragraph should be removed since no data collected by the authors corroborates the dominant symbiont of the mothers or larvae.

• Elder H, Million WC, Bartels E, Krediet CJ, Muller EM, Kenkel CD (2022) Long-term maintenance of a heterologous symbiont association in Acropora palmata on natural reefs. The ISME Journal

• Thornhill DJ, LaJeunesse TC, Kemp DW, Fitt WK, Schmidt GW (2006) Multi-year, seasonal genotypic surveys of coral-algal symbioses reveal prevalent stability or post-bleaching reversion. Marine Biology 148:711-722

4.3 Interventions for thermal tolerance:

“manipulation of coral-algal endosymbiosis”

~ the authors have not referenced the Symbiodiniaceae as “algae” yet in this manuscript or defined the use of “algae” to mean Symbiodiniaceae. For consistency in text and lack of confusion with readers this should be changed to reflect the terms used to reference the Symbiodiniaceae symbionts throughout the manuscript.

“Human interventions applying selective breeding in coral sexual propagation has been proposed as one of the viable options to maintain genetic diversity and increase resilience in restoration efforts (Barott et al., 2021; Epstein et al., 2003; van Oppen et al., 2015, 2017; NASEM, 2019; Hancock et al., 2021); however, feasibility to potentially scale up efforts remain limited and costly without full understanding of tradeoffs (Edwards et al., 2015; Chamberland et al., 2017).”

~ Need commas here like "Human interventions, such as selective breeding in coral sexual propagation, has been..."

~ The way this is written it sounds here like only thermally resistant colonies (NB colonies in this paper) should be considered for selective breeding, but thermal resilience is not the only trait corals need to survive anthropogenic climate change, which also causes increased disease prevalence and ocean acidification along with other factors. Since B adults have also been successful thus far, it is likely they have a different trait necessary for acclimating well to their home reefs, like disease resistance, greater trophic plasticity, or that they utilize different trade-offs to deal with thermal stress. The data here does not necessarily support selective breeding for thermal tolerance with individuals that do not bleach, but instead the preference for using eggs from NB parents when creating crosses following bleaching events. Therefore, the authors should consider making less definitive statements here.

Figures:

Figure 3: Reproductive output was higher in bleached colonies and had fewer abnormalities than NB colonies, how do the authors explain this in relation to their statements that the bleached colonies are less fit in the discussion?

Figure 4: Panel B should include x-axis labels for clarity. It is confusing to look at a separate panel for the axis labels, especially since only those two panels in the figure share labels.

Reviewer #2: PLOS ONE Review

General comments

Lenz and colleagues present results from experiments spanning multiple years to investigate the potential for multigenerational effects of thermal stress on the coral Montipora capitata. They found that while thermal stress impaired aspects of reproduction regardless of if a colony bleached or not, parental bleaching history was an important driver of offspring success. These findings supporting the concept that bleaching-resistant colonies could be targeted for selective breeding to produce offspring that to support the restoration of more thermally-tolerant coral populations. The authors do an excellent job of highlighting the importance of their findings and avoid overextending their conclusions beyond the scope of their study. Their findings are particularly timely given the increasing severity, prevalence, and mortality caused by thermal stress on coral reefs across the globe. I have relatively minor comments that are articulated below that I recommend addressing before the ms is published.

There are no line numbers on the submission, thus I have not included them below but did my best to describe where in the ms I am referring to by referencing paragraphs within sections.

Introduction: The introduction is well written and provides a thorough yet concise overview of how thermal stress can impact coral reproduction and population dynamics, while also highlighting the knowledge gap this paper addresses (intra- and intergenerational impacts of bleaching).

Paragraph (henceforth, P) 4: The sentence regarding high recruitment in Mo’orea after the 2016 thermal stress is incomplete/lacking details compared to the other examples. The examples presented in P3 and P4 provide clear links between thermal stress/bleaching and reproduction in corals. However, the Mo’orea example suggests that the minor 2016 bleaching event may have had the potential to affect reproductive output, but provides no concrete information that some corals were reproductively compromised. How it is currently written, one could conclude that the minor bleaching in 2016 did not impact reproduction. Suggest revising to better leverage this example for the introduction.

P5: Sentence with “~70% of reef corals on shallow reefs” – is there a reason that these corals are distinguished as reef corals? If not, remove to avoid confusion, or if there is an important distinction between reef corals and other corals, provide details for clarity.

Methods

P2: Do the authors have any bleaching history of the colonies used in this study during 2014? If so, that would be worthwhile to include here or as supplemental information.

P5: For quantification of larval survivorship – was this recorded as the number of larvae remaining in an individual well-plate, or did this take into account the number of settlers (which is needed to quantify larval survivorship)? For example, if a well-plate started with 10 larvae, and 5 had settled after 7 days and 5 remained swimming, was this recorded as 100% survivorship (5 out of the 5 larvae remaining) or 50% survivorship (5 of the initial 10 placed in the well-plate)? This should be clarified in the methods section.

P6: (second sentence) – revise for clarity. Mentions using a generalized mixed effects model twice in the same sentence, but is only referring to one model.

P7: The survivorship curve analysis includes days 23 and 27, but these days are not mentioned in the previous section regarding when the number of offspring alive were counted (days 6, 7, 28, 53, and 59).

P7: Was the effect of fragmentation included in any of the models to test if fragmenting impacted reproduction? Given the reported size of the fragments, there is not reason to believe there would be any impact, but this would be worth mentioning somewhere in the ms.

Results

Discussion: The discussion is well written and thorough. The authors did a nice job of placing their findings in the context of previous work (e.g. reduction in no. of eggs/bundle), proposing potential hypotheses for the patterns they observed (and supporting evidence), expanding the discussion to broader concepts of population ecology, and explaining how these results can be used to inform interventions like selective breeding.

Figures

Figure 2: Suggest choosing a color palette that facilitates distinguishing between B and NB colonies. I can understand why these were chosen to look like healthy (NB) vs. bleached (B) colonies, but the colors are difficult to quickly distinguish.

Figure 3: 3A -Lines are not appropriate for this type of plot – these data would be more clearly presented as a bar plot since each night of spawning is a discrete event, not continuous as the lines connecting the dots imply.

3B-E: suggest adding a legend for the B and NB colors even though it is on the x-axis labels (and hidden in the 2016 panel of Figure 3A.

Figure 4B: Add x-axis labels to this plot, it is difficult to follow them up from Fig 4C.

4C: Error bars on stacked bar plots make for very busy figures that are difficult to interpret and take away from the data being shown. Suggest individual bars for each of the developmental stages for each cross – this would make the difference in developmental stages easily observable by looking at Fig 4C.

4D-F: Similar to Fig 4, these colors are similar and thus hard to distinguish, especially given the that these are small panels within a 6-panel figure. Suggest selecting a different color scheme.

Reviewer #3: I enjoyed reading "Parental effects provide an opportunity for coral resilience following major bleaching events". I think the manuscript is sound and comprehensive look at colony fecundity, well-written, and almost ready for publication. My only reservation is some aspects of the discussion don't quite jive with the findings which I will detail below. However, I think this is good work and after a tidy up of some claims in the discussion, warrants publication. Despite my concerns with the discussion, I think the summary in the last paragraph "In our study... to balance reduced survivorship and settlement" to be a perfect summary of the findings.

Major concerns

*To me, the main result is that parental bleaching history did not significantly affect colony fecundity, the size of eggs, or eggs per bundle. If true, this runs counter to a lot of the discussion which emphasizes how bleaching affects reproductive capacity. For example on page 12, authors claims that this study "demonstrates the impact of bleaching on reproductive capacity" but the stated evidence doesn't show sig differences between B/NB (egg volume, eggs/bundle, abnormalities). Moreover, this carries over to the fertilization, survival, and settlement result where there seem to be no differences between larvae from B and NB colonies. On pg 15, authors claim that NB colonies have higher fitness, but I don't see how the results of this study support that for the reasons above plus seemingly minor effects on survival (I didn't see the magnitude of maternal effect on survival mentioned... seems small).

*Consider making maternal effects a larger focus of the discussion if this was the main sig result (and few outright B/NB differences).

*Discussion seems to be a bit fast and loose when describing insignificant effects. A focus on effect size could help support your claims. For example on pg 12, authors claim that 'maternal effects are demonstrated from fertilization through survival' but the maternal effect on fertilization is insignificant and the effect on survival seems to be quite small (Fig 4E inset). Same issue with "colonies have higher fitness" claim on pg 15.

*The sig. difference in proportion colonies spawned 2016 vs 2017 seems to be largely due to the third month, which is a light spawn in 2016 and no spawn in 2017. There seems to be little difference in months 1 and 2 between years. This may affect how the effects of 2015 bleaching are interpreted

*Where is there a significant maternal effect of NB/B? Do analyses need to be redesigned to test for maternal NB/B versus paternal NB/B?

Minor concerns

*Not sure if this is due to journal format, but Tables were not included in the submission which made it difficult to evaluate some claims

*I doubt it will change the conclusions, but technically, proportions and counts shouldn't use gaussian error distribution in linear mixed model

*pg11 Results section 3.2 p value 0.342 is in a different font

*Consider shrinking the discussion section on interventions

* Could be due to PDF conversion, but figures were blurry

* pg 12 please add 'for example' to describe what "delayed beneficial maternal effects might be" (you give more details later). maybe maternal effects need to be earlier in the disc. and a bigger part of the manuscript.

* What led to 2016 vs 2017 differences in abnormality?

* What could have led to B*B and NB*NB having faster development?

*Settlement rates seem a bit low. Is settlement typically <50% for this species?

6. PLOS authors have the option to publish the peer review history of their article (what does this mean?). If published, this will include your full peer review and any attached files.

Reviewer #1: No

Reviewer #2: No

Reviewer #3: No

---

## [Author Response · Author response to Decision Letter 0]

2 Apr 2024

Response 1: We have ensured that the submitted revisions follow the PlosONE style requirements and file naming as advised by the links provided above. 

“National Science Foundation Graduate Research Fellowship Program

Paul G. Allen Family Foundation.”

Response 2: The funders had no role in the design, execution, data collection, analysis decision to publish or preparation of the manuscript and will amend with the recommended language to include in our Funding Statement. See Response 3 below. 

If the funders had no role, please state: "The funders had no role in study design, data collection and analysis, decision to publish, or preparation of the manuscript."

“This work was supported by funding from Paul G. Allen Family Foundation to RG and the National Science Foundation Graduate Research Fellowships to EAL. We would like to thank members of the Gates Coral Laboratory for their technical support and advice, especially Jen Davidson and Dr. James Guest, we are grateful to Mary Hagedorn, Amy Moran, and Peter Marko for their feedback on the manuscript, the many volunteers especially Dyson Chee, Megan Buras, Katie Allen, Shayne Fabian, and Kat McPherson and the security Greg Miranda and Moses at Moku o Loʻe who ensured safety and the success of this research. We dedicate this research to Dr. Ruth D. Gates and her infectious enthusiasm that pushed human-assisted evolution to the forefront of coral biology – you rock!”

“National Science Foundation Graduate Research Fellowship Program

Paul G. Allen Family Foundation.”

Please include your amended statements within your cover letter; we will change the online submission form on your behalf of the coauthors:

Response 3: Thank you for the clarification, we have removed information regarding our funders from the acknowledgements and would like to include the following as our Funding Statement:

“This work was supported by funding from Paul G. Allen Family Foundation to RDG and the National Science Foundation Graduate Research Fellowships to EAL. The funders had no role in study design, data collection and analysis, decision to publish, or preparation of the manuscript.”

Response 4: Data repository is on GitHub

https://github.com/ealenz/Mcap-BNB-Reproduction-SelectiveBreeding

Response 5: The study did not require an IRB or ethics committee and this information is not relevant to the research that was conducted. 

6. We note that Figures 1B, 2, and 4A in your submission contain copyrighted images. All PLOS content is published under the Creative Commons Attribution License (CC BY 4.0), which means that the manuscript, images, and Supporting Information files will be freely available online, and any third party is permitted to access, download, copy, distribute, and use these materials in any way, even commercially, with proper attribution. For more information, see our copyright guidelines: http://journals.plos.org/plosone/s/licenses-and-copyright.

1. You may seek permission from the original copyright holder of Figures 1B, 2, and 4A to publish the content specifically under the CC BY 4.0 license.

Response 6: These images were produced and provided by the lead author of the paper. I, Elizabeth Lenz, give permission for these images to be used for the publication.

7. Please include your tables as part of your main manuscript and remove the individual files. Please note that supplementary tables should be uploaded as separate "supporting information" files.

Response 7: Thank you, we have included the tables as part of the main manuscript. 

Response 8: We reviewed our reference list and ensured it is updated.

Additional Editor Comments:

Message from the Editorial Office regarding a deceased co-author: For a deceased co-author, this would need to be indicated in the title page of the manuscript in addition to the acknowledgments. When submitting your manuscript, as the corresponding author you must accept the responsibility for the integrity of all data collected and analyzed by the deceased author. Suggested text for this in the Acknowledgements section is "[Author] passed away before the submission of the final version of this manuscript. [Corresponding author] accepts responsibility for the integrity and validity of the data collected and analyzed." Please also add a "†" symbol next to the author's name in the author list and include a note that this author is deceased.

Response 9: and have included the following: 

“Dr. Ruth D. Gates passed away before the submission of the final version of this manuscript. Dr. Elizabeth A. Lenz accepts responsibility for the integrity and validity of the data collected and analyzed." 

Please also add a "†" symbol next to the author's name in the author list and include a note that this author is deceased.

Reviewers' comments:

Reviewer's Responses to Questions

Comments to the Author

1. Is the manuscript technically sound, and does the data support the conclusions?

Reviewer #1: Partly

Reviewer #2: Yes

Reviewer #3: Partly

Response to Reviewers: We appreciate the honest response by the reviewers and hope we have fully addressed the concerns.

2. Has the statistical analysis been performed appropriately and rigorously? 

Reviewer #1: Yes

Reviewer #2: Yes

Reviewer #3: Yes

Response to Reviewers: We appreciate the positive response by the reviewers and appreciate the acknowledgement of our work in our statistical analyses of the study.

3. Have the authors made all data underlying the findings in their manuscript fully available?

Reviewer #1: Yes

Reviewer #2: Yes

Reviewer #3: Yes

Response: We appreciate the positive responses from the reviewers in meeting the PLOS data policies. 

4. Is the manuscript presented in an intelligible fashion and written in standard English?

Reviewer #1: Yes

Reviewer #2: Yes

Reviewer #3: Yes

Response: We are grateful to the reviewers for taking the time to copy edit the manuscript and ensure the contents meets the PLOS ONE requirements. 

5. Review Comments to the Author

Reviewer #1: Reviewer comments on lines following "~".

Direct quotes being addressed from manuscript are in "".

Review comments are broken up by sections and sub-headers in sections.

Response: Thank you for providing clear instructions and guidance on the feedback provided. We greatly appreciate the clarity and thoughtfulness behind the comments.

Introduction:

“Coral bleaching is known to impact sexual reproduction (Baird & Marshall, 2002; Fisch et al.,2019) and recruitment (Hughes et al. 2019; Price et al., 2019)… colonies that bleached and recovered (Ward et al., 2002).”

~ This reads like regurgitated info. I recommend adding a concluding sentence to this paragraph that highlights what you want the reader to understand about the importance of your project from the inclusion of this paragraph in the intro.

Response: We appreciate the feedback on providing a stronger concluding sentence to help contextualize the study for the readers and provide stronger understanding.

LN 81-93: “Previous studies have identified some of the way coral bleaching can impact aspects of sexual reproduction (Baird & Marshall, 2002; Fisch et al., 2019) and dampen recruitment (Hughes et al. 2019; Price et al., 2019). For example, after the 1987 coral bleaching event in the Caribbean, Orbicella annularis recovered from bleaching by metabolizing tissue biomass, but did not complete gametogenesis in the following months, whereas colonies that had not bleached of the same species were able to develop and release gametes (Szmant & Gassman, 1990). Similarly, during the 1998 bleaching event on the Great Barrier Reef, bleached corals showed high variation in reproduction compared to colonies resistant to bleaching nearby that experienced the same thermal stress. For acroporid species, reproductive polyps were more common in colonies that did not bleach, with larger eggs at higher densities per polyp than colonies that bleached and recovered (Ward et al., 2002). More resolution is needed to better understand the impact and extent of coral bleaching events on the early life cycles of coral, from the stress event through recruitment.”

~ There is differential bleaching observed dramatically across individuals so close to each other on the same reef. Has microsatellite work been done on these colonies? Is there a possibility of cryptic species/lineages across the bleaching susceptible and resistant corals here?

Response: We did not conduct microsatellites on these colonies and cannot conclude a pattern of cryptic species/lineages within the samples. However, there is evidence that colonies more vulnerable to bleaching have clade C symbionts and that corals more resistant to bleaching are dominated by thermotolerant clade D (Cunning et al. 2016). Since 2014, former members of the Gates Coral Lab, including Drs. Raphael Ritson Williams, Christopher Wall, Hollie Putnman, Ross Cunning, Shayle Matsuda, Katie Barott and now the Coral Resilience Lab at the Hawaiʻi Institute of Marine Biology continue to conduct studies to understand and explain the mechanisms that drives the differential bleaching responses to marine heatwaves in Kāneʻohe Bay. This mechanism behind this differential bleaching phenomenon remains unknown.

~ The authors talk a lot in the intro about how bleached individuals do not produce as many eggs as individuals that did not bleach for several coral species, but the authors findings (and included previous work on M. capitata) do not support this. Why include so many examples of this information in the intro then? Maybe just say concisely in discussion.

Response: The intent of the introduction was to provide a thorough overview of what has been observed in the past, particularly with regards to corals in the acroporidae family. We believe it is worthwhile to keep as is and that a hypothesis for the difference in results could be attributed to acclimation over the last decade as waters have warmed.

“This 

---

## [Decision Letter · Decision Letter 1]

5 Jun 2024

PONE-D-23-24872R1Parental effects provide an opportunity for coral resilience following major bleaching eventsPLOS ONE

Dear Dr. Lenz,

Thank you for submitting your manuscript to PLOS ONE. After careful consideration, we feel that it has merit but does not fully meet PLOS ONE’s publication criteria as it currently stands. Therefore, we invite you to submit a revised version of the manuscript that addresses the points raised during the review process. Please see the comments below from one of the reviewers.

We look forward to receiving your revised manuscript.

Kind regards,

Atsushi Fujimura

Academic Editor

PLOS ONE

Journal Requirements:

Reviewers' comments:

Reviewer's Responses to Questions

**Comments to the Author**

1. If the authors have adequately addressed your comments raised in a previous round of review and you feel that this manuscript is now acceptable for publication, you may indicate that here to bypass the “Comments to the Author” section, enter your conflict of interest statement in the “Confidential to Editor” section, and submit your "Accept" recommendation.

Reviewer #3: (No Response)

2. Is the manuscript technically sound, and do the data support the conclusions?

Reviewer #3: Yes

3. Has the statistical analysis been performed appropriately and rigorously? 

Reviewer #3: Yes

4. Have the authors made all data underlying the findings in their manuscript fully available?

Reviewer #3: Yes

5. Is the manuscript presented in an intelligible fashion and written in standard English?

Reviewer #3: Yes

6. Review Comments to the Author

Reviewer #3: Maybe they were lost in the format of my response, but several comments were not addressed in the authors response. I list them here as suggestions that may improve the manuscript.

Major concerns:

*Discussion seems to be a bit fast and loose when describing insignificant effects. A focus on effect size could help support your claims. For example on pg 12, authors claim that 'maternal effects are demonstrated from fertilization through survival' but the maternal effect on fertilization is insignificant and the effect on survival seems to be quite small (Fig 4E inset).

*The sig. difference in proportion colonies spawned 2016 vs 2017 seems to be largely due to the third month, which is a light spawn in 2016 and no spawn in 2017. There seems to be little difference in months 1 and 2 between years. This may affect how the effects of 2015 bleaching are interpreted

*Where is there a significant maternal effect of NB/B? Do analyses need to be redesigned to test for maternal NB/B versus paternal NB/B?

Minor concerns:

*I doubt it will change the conclusions, but technically, proportions and counts shouldn't use gaussian error distribution in linear mixed model

7. PLOS authors have the option to publish the peer review history of their article (what does this mean?). If published, this will include your full peer review and any attached files.

Reviewer #3: No

---

## [Author Response · Author response to Decision Letter 1]

3 Sep 2024

Reviewer #3: Maybe they were lost in the format of my response, but several comments were not addressed in the authors response. I list them here as suggestions that may improve the manuscript.

RESPONSE: We apologize for missing reviewer 3 comments in the first round of our response to reviewers. We appreciate the opportunity to address them as we had not seen Reviewer 3’s comments prior.

Major concerns:

*Discussion seems to be a bit fast and loose when describing insignificant effects. A focus on effect size could help support your claims. For example on pg 12, authors claim that 'maternal effects are demonstrated from fertilization through survival' but the maternal effect on fertilization is insignificant and the effect on survival seems to be quite small (Fig 4E inset).

RESPONSE: We appreciate the comment and in response we have revised the results and discussion. We have added the following language to the results: 

LN 328-338: “While reproduction continued in the colonies examined, we found that cross-type did have an effect on fertilization, embryonic development, and percent larval survivorship (Figure 4; Table 2). Specifically, fertilization success in the NB × NB cross-type was higher than the B × NB and NB × B cross-types (Figure 4B; Table 2; post-hoc P = 0.002 and 0.010, respectively) but not B × B (post-hoc P = 0.163). The fertilization success in cross-type B × B also did not differ between NB × B (post-hoc P = 0.250), but was higher than B × NB (post-hoc P = 0.047).. Cell division advanced beyond the 2-cell stage more quickly for within cross-types (B × B and NB × NB) than between cross-types (B × NB and NB × B) at 3-h post-fertilization. Embryos from both B × B and NB × NB cross-types reached the 16-cell stage at 3-h post fertilization, whereas embryos from B×NB and NB × B crosses developed at a slower rate and only reached the 4-cell stage (Figure 4C, Table 2).”

We also added the following to the Discussion:

LN 420-422” “We demonstrate parental effects, or cross-generational plasticity, in M. capitata, with parent cross-type having an effect on fertilization and embryonic development with maternal effects apparent in offspring survivorship.”

*The sig. difference in proportion colonies spawned 2016 vs 2017 seems to be largely due to the third month, which is a light spawn in 2016 and no spawn in 2017. There seems to be little difference in months 1 and 2 between years. This may affect how the effects of 2015 bleaching are interpreted

RESPONSE: We understand the reviewer’s concern that there is little difference between the first two months of the years compared and impact may be driven by the lack of spawning in the third month. We have added the following to the results: 

LN 307-311: “The proportion of colonies releasing gametes significantly differed by year (P < 0.001) which may largely be due to some spawning in 2016 compared to no spawning in 2017 during the third month (August). In 2017, the proportion of colonies participating in spawning events was 36% lower than in 2016. In both years, the second month of the spawning season had the highest proportion of colonies spawning.”

*Where is there a significant maternal effect of NB/B? Do analyses need to be redesigned to test for maternal NB/B versus paternal NB/B?

RESPONSE: Montipora capitata are hermaphroditic colonies, see Figure 2B for matrix of crosses. We identified an effect of NB and B for egg donors in the larval survivorship in the Summary of Cox proportional hazards see Figure 4F. We found that larval survivorship was influenced by the bleaching status of the egg donors. Offspring from non-bleached (NB) corals exhibited higher survival rates.

Minor concerns:

*I doubt it will change the conclusions, but technically, proportions and counts shouldn't use gaussian error distribution in linear mixed model

RESPONSE: We greatly appreciate the reviewer pointing out this error in our analysis and we have revised the methods to apply glmer to the proportion and count data. Where assumptions of normality were not met, we applied the Kruskal-Wallace test for the embryonic development variables. Please see our revised methods, results, and tables.

LN 265-297: “All analyses were conducted in R (R Core Team, 2014; v. 3.5.1). We used a generalized linear mixed effects model to determine the effects of bleaching history on spawning activity, number of eggs per bundle, and egg abnormality of the 8 B and 7 NB parental colonies observed (glmer in lme4: Bates et al. 2015) with a binomial (spawn/no spawn and proportion of abnormal eggs) and poisson (eggs per bundle count) response. Bleaching history (B/NB) and year (2016/2017) were included as fixed effects, and spawning month (1/2/3) and colony ID were included as random effects. To analyze total reproductive output and egg size, we used linear mixed effects models (lme in lme4: Bates et al. 2015) with bleaching history and year as fixed effects, and colony ID as a random effect. Analysis of variance (ANOVA) tables were generated using type II sum of squares (Anova in car: Fox and Weisberg 2011).Post-hoc analyses were conducted to further explore significant main effects and interactions. We utilized the emmeans package (Lenth, 2020) to calculate and compare the estimated marginal means (EMMs), which represent the predicted means of the response variable for each level of the fixed effects, adjusted for the other covariates in the model. Pairwise comparisons between the levels of the fixed effects were then performed using Tukey’s Honest Significant Difference (HSD) test to adjust for multiple comparisons. This approach allowed us to identify significant differences between specific treatment groups, while accounting for the variability associated with random effects.

To test the effects of parental bleaching history on offspring performance, we first analyzed the proportion of eggs fertilized using generalized linear mixed effects models with cross-type as a fixed effect and the egg donor and sperm donor as random effects. The proportion of eggs reaching each developmental stage (2-cell, 4-cell, 8-cell, and 16-cell), the Kruskal-Wallace test was applied as the dataset did not meet the assumption of normality. For post-hoc analysis, we performed the Dunn’s test for multiple pairwise comparisons to determine which specific cross-types differed. To analyze the proportion of larvae that settled at 7, 28, and 59-days post-fertilization, we used a generalized linear mixed effects model with cross-type and day (7, 28, and 59-d post-fertilization) as fixed effects and colony ID of egg donor and sperm donor as random effects. Lastly, we generated survivorship estimate curves to visualize offspring fate by cross-type with ggsurvplot of the census over time (i.e., days 6, 7, 23, 27, 28, 53, and 59 post-fertilization) (survfit in survminer; Kassambara et al., 2017). Cox proportional hazards (CPH) model was used to analyze the effects of cross, egg donor, and sperm donor individually on offspring survivorship (coxph in survminer; Kassambara et al., 2017). Dispersion parameters were inspected through a simulation-based approach (DHARMa package: Hartig, 2019).”

---

## [Editor Report · Decision Letter 2]

18 Sep 2024

Parental effects provide an opportunity for coral resilience following major bleaching events

PONE-D-23-24872R2

Dear Dr. Lenz,

We’re pleased to inform you that your manuscript has been judged scientifically suitable for publication and will be formally accepted for publication once it meets all outstanding technical requirements.

Kind regards,

Atsushi Fujimura

Academic Editor

PLOS ONE

---

## [Editor Report · Acceptance letter]

26 Sep 2024

PONE-D-23-24872R2 

PLOS ONE

Dear Dr. Lenz, 

I'm pleased to inform you that your manuscript has been deemed suitable for publication in PLOS ONE. Congratulations! Your manuscript is now being handed over to our production team.

Kind regards, 

on behalf of

Dr. Atsushi Fujimura 

Academic Editor

PLOS ONE